# Community health extension workers' training and supervision in Ethiopia: Exploring impact and implementation challenges for non-communicable disease service delivery

**Azeb Gebresilassie Tesema**[1,2]*, **David Peiris**[1], **Seye Abimbola**[1,3], **Whenayon S. Ajisegiri**[1], **Padmanesan Narasimhan**[4], **Afework Mulugeta**[2], **Rohina Joshi**[4,5]

1 The George Institute for Global Health, University of New South Wales (UNSW), Sydney, Australia, 2 School of Public Health, Mekelle University, Mek'ele, Ethiopia, 3 School of Public Health, University of Sydney, Sydney, Australia, 4 School of Population Health, University of New South Wales, Sydney, Australia, 5 The George Institute for Global Health, New Delhi, India

* azeb18@gmail.com, atesema@georgeinstitute.org.au

**Data Availability Statement:** All relevant data contributing to the findings are within the paper.

## Abstract

Training and supervision of health workers are critical components of any health system; thus, we assessed how they impact health extension workers' (HEWs) role in non-communicable disease (NCD) service delivery in Ethiopia's health extension program (HEP), using an in-depth qualitative study conducted in 2019.The study covered two regions—the Tigray and the South Nations, Nationalities and Peoples Region (SNNPR)—and involved the Federal Ministry of Health. We conducted twenty-seven key informant interviews with federal and regional policymakers, district health officials, health centre representatives and HEWs. Participants highlighted substantial implementation challenges with training and supervision practices delivered via the HEP. Training for NCDs lacked breadth and depth. IT was described as inconsistently delivered with variable availability within and between regions; and when available, the quality was low with scant content specific to NCDs. HEP supervision was inconsistent and, rather than being supportive, mainly focused on finding faults in HEW work practices. Supervisors themselves had skill gaps in critical areas overall, and specifically concerning NCDs. HEWs' performance appraisal encompassed too many indicators, leading to excessive complexity, which was burdensome to HEWs. This, negatively impacted HEW motivation and compromised service delivery. HEW involvement in non-HEP activities (such as promoting other government programs) often competed with their core mandates, thus affecting HEP service delivery.Efforts to address training and supervision constraints in Ethiopia's HEP should focus on improving the quality of NCD training for HEWs and supervisors, shifting from authoritative to supportive supervision, simplifying performance appraisal and reducing competing attention from other programs.

Raw data contributing to the analyses in this study are stored on a secure network and not publicly available in order to protect participant confidentiality and to adhere to the University of New South Wales (UNSW) ethics policy.

**Funding:** The project was supported by the George Institute for Global Health, Australia through the Seed Grant fund dedicated for under-served populations in low middle income countries for 2019/2020. The UNSW Scientia Scholarship program supports AGT and WA. The UNSW Australian Government Research Training Program Scholarship supports AGT. SA was supported by the Australian National Health and Medical Research Council (NHMRC) through an Overseas Early Career Fellowship (APP1139631). RJ is supported by the Australian National Heart Foundation (APP 102059) and a UNSW Scientia Fellowship. DP is support by NHMRC career Development Fellowship, Level 2 and Australia National Heart Foundation Future Leader Fellow. The funders had no role in study design, data collection and analysis, decision to publish, or preparation of the manuscript.

**Competing interests:** The authors have declared that no competing interests exist.

# Background

The shortage of a skilled health workforce and the rising morbidity and mortality from non-communicable diseases (NCDs) in resource-limited settings, including Ethiopia, have made community health worker (CHW) programs an essential strategy to address the growing NCD burden [1]. Recent studies in Bangladesh, Nepal and Uganda have demonstrated the critical role of CHWs in a wide range of NCD services, including health promotion, counselling, screening, provisional diagnosis, management of prescribed medication adherence, and referral [2–4].

Ethiopia introduced a Health Extension Program (HEP) in 2003 to deliver health promotion, disease prevention and selected curative health services at the community and household levels [5, 6]. It was developed to address health system weaknesses, disparities in health outcomes between urban and rural populations, and a critical shortage of skilled health workers [7–9]. Initially, the program focused on 16 HEP packages, covering maternal and child health and infectious diseases and was subsequently revitalised in 2016/17 to include NCD services. The inclusion of NCDs in the HEP expanded the role of health extension workers (HEWs) [6, 10–14]. Under the new program, their tasks included community health education on NCDs, raising awareness of NCD risk factors, screening for early NCDs detection, and referring patients with more complex needs to health centres. See Box 1 for historical context and details on Ethiopia's HEP.

---

## Box 1. The context: The Ethiopian health extension program

The primary goal of the HEP is to improve access to health care, particularly for under-served populations. The stimulus for its creation was severe health system constraints characterised by health worker shortage, maldistribution of health resources and limited-service coverage in rural regions [20]. Initially, the program focused on agrarian communities and 16 preventive and promotive packages. Specifically, the programs covered family health, disease prevention and control, hygiene and environmental sanitation, first aid, and health education and communication [5, 6]. However, the number of packages and target communities have expanded over time, from solely promotive and preventive services to a more comprehensive package including selected curative services (**Fig 1**). In 2009, the urban HEP was launched to address the health challenges in urban areas. The main functions of the urban HEP package were ensuring family and environmental health; addressing communicable and NCDs; and providing first aid; and preventing accidents [21]. NCDs and mental health were introduced in rural areas in 2016/17 as part of the newly introduced second-generation HEP [6, 13].

*HEW role*: HEWs (as they are known in rural areas) and urban health extension professionals (UHE-ps) (as they are known in urban settings) are central to the implementation of the HEP. At first, they conducted outreach activities and home-to-home visits in the respective communities to support approved packages, particularly those related to maternal and child care services [13, 22–24]. Following implementation of the optimised HEP, and the inclusion of NCDs, the scope of work for HEWs has expanded to the prevention, promotion and early detection or screening of five major NCDs: cancer, diabetics, hypertension, cardiovascular diseases and chronic obstructive pulmonary disease, including Asthma [11, 25]. Broadly, their role includes: (1) awareness creation on NCDs and their risk factors, (2) conducting health education on harmful practices (such as imbibing alcohol) and promoting prevention activities (3) undertaking basic community

screening for early detection of NCDs (such as high blood pressure measurement and monitoring), (4) referring suspected cases to health centres, and (5) collecting and interpreting NCD data in the community [6, 10–12]. As we show in the results section, their actual role is rarely aligned with the full range of activities they are meant to undertake as they are currently focused only on a limited set of preventative services.

At the time of our fieldwork in 2019, Ethiopia had a functioning HEP, supported by 38,868 HEWs and 17,587 health posts. Around 85% of HEWs were deployed in rural health posts. Health posts represent the service delivery point closest to the community in rural settings; on average, services 3,000–5,000 people [5, 6].

***HEW selection and training***: All HEWs have formal educational qualifications: graduation from 10th grade in rural settings and a clinical nursing diploma in urban areas. They are selected to serve in the HEP according to national criteria that include residence in the village, capacity to speak the local language, and willingness to remain in the village to serve the community. After recruitment, HEWs and UHE-ps attend pre-service training on the HEP and the package of services they deliver to the community. For those with Grade 10- level education, the training is of 12 months duration, includes practical placements in health facilities (i.e. health centres) and leads to a Level III certificate qualification upon completion. The pre-service training is limited to three months for those with a clinical nursing diploma [6, 23, 26]. After graduation, both HEWs and the UHE-ps are deployed as salaried government employees.

HEWs can earn a Level IV qualification (diploma level certification), on completion of an additional year of training after obtaining a Level III qualification [11, 14, 27]. In 2020, around 25% of HEWs had completed Level IV training. A more structured in-service training scheme called integrated refresher training (IRT) program, is also in place to address the knowledge and skill gaps needed to implement the HEP and supplement the pre-service training [6]. More recently, the country has implemented a curriculum offering further opportunities for HEWs to pursue a degree program in family health.

***HEW supervision***: HEWs receive supervision from primary health care (PHC) units, supported by a supervisory framework involving federal, regional, and *woreda* (district) health offices. As per the framework, the Federal Ministry of Health (FMoH) is mandated to provide supportive supervision to regional, district and selected PHC units, including health posts in collaboration with the regional health bureau. The regional health bureau (RHB) has a supervisory responsibility over districts and PHC facilities, while *woreda* health offices are mandated to provide an integrated supportive supervision to all PHC units under their jurisdictions. RHBs undertake supervisory visits twice a year while *woreda* health offices conduct theirs quarterly. At the *woreda* level, the supervisory team consists of a health officer, public health nurse, environmental/hygiene expert, and a health education expert. Team members are oriented in the use of various tools and methods (such as peer review and performance assessment tools) and have a detailed schedule for each supervisory visit. Supervision focuses on overall program control, including planning, implementation and monitoring and evaluation [6–8, 26, 28–30].

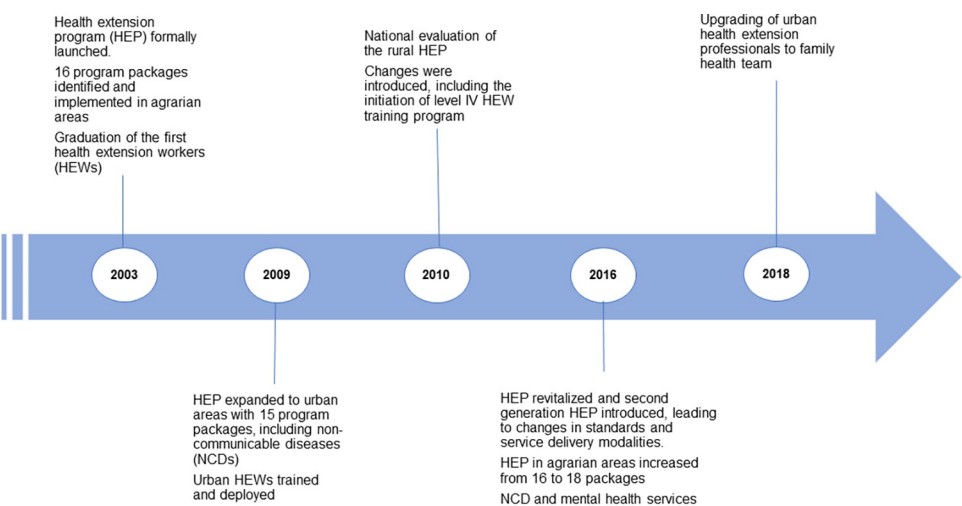

**Fig 1. Major HEP and Ethiopian health system development timeline.** Source: Ethiopia Federal Ministry of Health roadmap for optimizing the Ethiopian health extension program 2020–2035 [6].

Improving NCD service uptake as part of the expanded HEP requires new HEW capacity strengthening practices, that are well integrated with existing non-NCD packages [6, 15]. Studies in Ethiopia and elsewhere highlight the importance of training and supervision to enhance CHWs' capacity, performance, and motivation [6, 15–17]. Training is an integral component of CHW programs in many settings and is often a prerequisite for CHWs to become and remain effective in the health system [17]. Similarly, supportive supervision makes the health system more functional. It identifies and resolves problems, strengthens relationships, optimises resource allocation, improves health worker performance, and sustains job satisfaction [18]. The World Health Organization (WHO) also recommends continuously applying performance assessment and using community feedback to enhance supervision quality [19].

Furthermore, studies suggest linking supervision and performance appraisal as this helps to incentivise and motivate CHWs and strengthen health service delivery. However, despite the wide recognition and substantial evidence on the importance of training, supervision and performance appraisal, the available evidence has not been sufficiently granular and has been uneven across and within countries [19]. In addition, because NCDs are lifelong diseases, often with a different population focus to non-NCD programs, the workforce support requirements for NCD program integration need to be closely investigated, identifying needs that are specific to NCDs and those that overlap with other programs.There has been limited research on how best to increase HEW capacity for NCD care in Ethiopia. In this study we aimed to understand how NCD supervision, training, and performance appraisal processes are being implemented as part of the HEP and to identify enablers and constraints to NCD program implementation.

## Methods

### Study design

We conducted an in-depth qualitative study with a diverse sample of HEWs, UHE-ps, supervisors, and *woreda*, regional and FMoH officials from August to October 2019.

## Study setting

Ethiopia is divided into nine regions and two city administrations (i.e. Addis Ababa and Dire Dawa). The regions are divided into zones, each of which is sub divided into lower administrative units called *woredas*, which are the primary administrative units in the country [31]. Each *woreda* is subdivided into the lowest administrative units at the village level, called kebele, which constitute around 5,000 inhabitants served by a health post [32].

The study was conducted in Tigray Region and the South Nations, Nationalities and Peoples Region (SNNPR) withrespondents also from the FMoH, Addis Ababa. The two regions were purposely selected based on advice from FMoH that these regions were participating in the second-generation HEP rolloutThe sample from Tigray included three *woredas* (Atsebi-Wemberta, Hawzien and KilteAwla'elo and Mekelle Special Zone).

Similarly, three *woredas* from SNNPR were included (Hawassa city; Kachabira; and Hadaro Tunto Zura). The *woredas* in both regions were selected in consultation with the respective regional health bureau representatives. We also targeted health posts that had introduced the second-generation HEP (i.e. the revitalised HEP that included NCDs as additional packages).

## Participant recruitment and data collection

HEWs from the above areas were purposively invited to participate in interviews based on their experience and involvement in delivering the second-generation HEP. Contact details for HEWs were obtained from the catchment health centre. Key informant interviews were also conducted with national and regional policymakers selected from the FMoH, the two RHBs, *woreda* health officers and health centre managers (**Table 1**). Administrative officials were contacted initially via emails or personal visits to their offices, and interviews were conducted in person. HEWs and health centre staff were approached after permission was granted by the *woreda* management to visit the health facilities. Twenty-seven participants were interviewed in total.

Interview guides were developed after reviewing various national strategic documents and literature from other settings, and were iteratively refined as interviews took place (please see S1 File). Questions focussed on health system responsiveness to the rising NCD burden, the role of HEWs in addressing NCDs, and the level of training and supervision provided. The primary investigator (PI, AGT) and local research assistants recorded the interviews using a combination of audio recording and note-taking. The PI, a research team member (AM) and the research assistants met on an ongoing basis to discuss new identified data and revise the data collection tools, when necessary, for subsequent interviews.

This process continued until data saturation was achieved and no substantively new themes of interest were identified [33]. The interviews were conducted face-to-face using official working languages—Amharic and Tigrigna—and each interview lasted for around one hour. The interviews were held in the place and time of each interviewee's choice to avoid interference with the regular duty of participants and ensure the interviews were held in a private space.

The resulting data were transcribed into English by local research assistants and the PI. The PI reviewed all the transcripts independently to check for errors by listening back to the audio-recording and reading the transcripts simultaneously. The PI also checked transcription and translation accuracy by transcribing and translating three interviews. The transcriptions were further supplemented with notes/memos during data collection and discussion with the research assistants. Data familiarisation was done by reading and making notes prior to data coding. Additional material on the methods is available elsewhere [32].

**Table 1. Characteristics of study participants, Ethiopia, 2022.**

| Participant(P) code | Region/location | Study participant's working area/work |
|---|---|---|
| Participant 1 (P1) | Addis Ababa | Federal Ministry of Health (FMoH) |
| P2 | Addis Ababa | FMoH |
| P3 | Addis Ababa | FMoH |
| P4 | South Nations, Nationalities and Peoples Region (SNNPR) | Urban health extension professional (UHE-p) |
| P5 | SNNPR | Health extension worker (HEW) |
| P6 | SNNPR | UHE-p |
| P7 | SNNPR | HEW |
| P8 | SNNPR | HEW |
| P 9 | SNNPR | Health centre |
| P10 | SNNPR | Health centre |
| P11 | SNNPR | *Woreda* health office |
| P12 | SNNPR | *Woreda* health office |
| P13 | SNNPR | Regional health bureau (RHB) |
| P14 | SNNPR | RHB |
| P15 | Tigray Regional State | HEW |
| P16 | Tigray Regional State | HEW |
| P17 | Tigray Regional State | HEW |
| P18 | Tigray Regional State | HEW |
| P19 | Tigray Regional State | UHE-ps |
| P20 | Tigray Regional State | UHE-ps |
| P21 | Tigray Regional State | Health centre |
| P22 | Tigray Regional State | Health centre |
| P23 | Tigray Regional State | Health extension program supervisor |
| P24 | Tigray Regional State | *Woreda* health office |
| P25 | Tigray Regional State | *Woreda* health office |
| P26 | Tigray Regional State | RHB |
| P27 | Tigray Regional State | RHB |

## Data analysis

We coded the verbatim transcripts using NVivo 12 software (QSR International, Vic). The PI performed the data coding and overall analysis. One of the research team members (RJ) independently coded a sample of three interviews. Both the PI and RJ discussed and refined the codes with input from the rest of the research team. The coding framework was informed by the broader CHW performance framework developed by Agarwal et al. and built on our previous findings [34]. We used as many codes as possible during the initial coding process, and later combined those with similar themes into categories. Candidate themes identified from the analyses were indexed under three focus areas of interest—HEW training, supervision and performance appraisal for NCD delivery. A well-functioning health system requires integrated,comprehensive and dynamic training and supervision practices. However, supervision and training are meaningful only within the context of the specific roles played by health workers [6, 15–17]. Consequently, we asked participants about HEWs current roles vis-à-vis their envisioned or anticipated role in delivering NCD services within the HEP.

Thematic content analysis was used to analyse the data. We generated new themes both inductively—that is, from the experiences and views of the participants—and deductively— that is, from previous findings and our own experiences. The team held discussions during the

coding process of interview transcripts, development of the coding framework and interpretation and writing up of identified themes. We reported the results in accordance with the consolidated criteria for reporting qualitative research (COREQ) (please see S2 File).

## Ethical considerations

Ethics approval for the study was granted by both the Ethiopian Public Health Institute, National Research and Research Ethics Office (EPHI-IRB-194-2019), Ethiopia, and the University of New South Wales (UNSW) Human Research Ethics Committee (HC190014) Sydney, Australia. Research support letters were obtained from the FMoH, the two RHBs and each *woreda* administration. Respondents were informed about the study's aim and were included in the study once their written consent was obtained. All invited participants agreed to participate in the study. All interviews were conducted in a quiet and private place chosen by the participant. To ensure confidentiality and anonymity of the data, we replaced participants' names with codes during data analysis and presentation (**Table 1**).

## Results

### The context: Health extension worker role in delivering non-communicable disease services

Our findings showed that although the scope of work for HEWs in the newly revised HEP encompasses the prevention, promotion, early screening of NCDs and referral of suspected cases to PHC units, their role was generally limited to NCD prevention services. Notably, HEW participants stressed that their services focus on creating awareness on diet, and educating the community about the importance of physical activity, salt reduction, weight control and tobacco cessation. *'Our focus is on educating the community about NCD risk factors and some mitigating measures'*, reported a *woreda* health office participant from SNNPR (P12). However, the NCD prevention activities were not structured and not provided on a regular basis as for other services. A HEW participant from Tigray (P16) indicated that:

> While we have detailed lesson plan for other diseases, covering information on the nature of the disease, its symptoms, signs and medication, for NCDs (eg. diabetes), we provide only just general information such as not to take much salt.

A RHB participant from SNNPR (P14) concurred with the HEW:

> At the moment, the scope of work for HEWs is limited mainly to communicable diseases. Although there is a mandate to expand their role and provide NCD services, but we didn't do a good job.

However, a participant from the FMoH (P1) had a different opinion:

> There are challenges, no doubt, but this doesn't mean that there are no NCD-related services. For instance, HEWs provide health education during their house-to-house visits, and in some places, they also measure blood pressure.

A *woreda* health office participant from Tigray (P24) added that, 'while they [HEWs] give counselling and health education services on preventing NCDs, these services are irregular'.

According to some participants, HEWs also provided screening for high blood pressure in a few areas, as the appropriate equipment was already available for maternal health activities.

However, services for screening other NCDs, such as diabetes, were not available. Participants attributed the absence of NCD services at the health post level to a lack or shortage of equipment, such as blood pressure machines and glucometers to screen for blood sugar level.'HEWs don't conduct screening for NCDs due to their [HEWs] skill gap, but importantly because of lack of equipment' noted a *woreda* health officer from SNNPR (P12). However, HEWs participants, particularly UHE-ps, explained that they assisted people with confirmed NCD cases and advised them to adhere to medications during their house-to-house visits.

Furthermore, participants noted that HEWs generally recorded and reported NCDs, but their involvement in NCD-related health information system was limited and not as structured as other programs. 'Unlike family planning or other maternal health services reporting, the HEWs don't regularly and seriously document and report NCD data to health centre', a *woreda* health office participant (P24) explained.

Participants further explained that HEWs' roles in referring NCD suspected cases to PHC units was almost non-existent as patients often bypassed the HEWs and made their 1st contact at the health centre level. As one HEW participant from SNNPR (P5) noted, 'We [HEWs] do not usually engage in the NCD referral process. We don't have a relevant referral form for NCDs even.'Participants mentioned that UHE-ps took on relatively more NCD roles than their counterparts in rural areas. This was because UHE-ps had better skills given all are trained nurses. It may also reflect the fact that NCD packages have been incorporated as part of the urban HEP since the program's establishment. Other enabling factors include the proximity of the UHE-Ps to health centres and a greater level of NCD awareness among urban residents compared to rural counterparts. A RHB participant from Tigray (P27) explained the role of UHE-ps as follows:

> The UHE-ps teach the urban residents about NCDs and associated risk factors through their house-to-house visits. She [HEW] has a blood pressure measurement apparatus for screening and immediately refers the person to a health centre when she detects a suspected case.

Within the context of these existing roles of HEWs, our findings on training, supervision and performance appraisal are organised into three themes.

### Theme 1: Formal and in-service training for health extension workers: Enablers and implementation challenges for non-communicable disease service delivery

**Subtheme 1.1 Pre-service training as an instrument of capacity building.** Some participants indicated that the Level IV training program had helped HEWs acquire new knowledge and skills on various aspects of the HEP. A *woreda* health office participant from Tigray explained that, 'HEWs with Level IV training are better equipped to understand and respond to the community's health problems than the Level III HEWs'. (P24). A HEW participant from SNNPR concurred, stating, 'the Level IV training helped me acquire good knowledge and skills that we didn't have before'. (P5). However, several participants identified bottlenecks during training, including limited availability of teaching aids and insufficient practical lessons, which ultimately affected the quality of the pre-service training and the readiness and capabilities of the trainees to provide the required services.

**Subtheme 1.2 Training without role progression.** Study participants repeatedly mentioned the absence of a conducive work environment in their post-training role as a potential barrier to delivering HEP packages, including NCD services. Theseincluded task

misalignment, inadequate provision of supplies and unmet expectations on the part of HEWs. The head of a *woreda* health office from SNNPR noted that, 'While some HEWs had the opportunity to upgrade to Level-IV qualification recently in our area, they are doing what they had been doing before'. (P12). A HEW from Tigray complained:'Even after completing the training, what is the benefit because we get placed in the same health post and do the same task and play the same role as we were in Level III. (P16).

However, a high-level regional policymaker (P26) pointed out that the issue was not about training or role misalignment but differing expectations of HEW roles. He argued that aligning expectations and providing sufficient resources to carry out their expanded role was challenging.

> Some Level IV HEWs believe their new role should be providing only curative services. This confusion creates disappointment as they get placed in the health post; in most cases the same facility they previously served.

However, a participant from the FMoH (P3) indicated that following their Level IV training, HEWs should have assumed an advanced role within the HEP, at least for NCD service delivery:

> During Level IV training, the HEWs are exposed to NCDs. Their role should have focused on awareness creation and screening for major NCDs, linking suspected cases with health centres for additional diagnosis and care.

**Subtheme 1.3 In-service training has benefits but is inadequate.** Participants acknowledged the importance of short-term in-service training and noted that currently, such opportunities were part of the IRT and available to HEWs at all levels. However, the ad hoc nature of this mode of training had led to substantial variation in uptake and not all regions and *woredas* benefitted from these opportunities. 'I can not say there is refresher training in our area. I attended only once since I started working here ten years ago', aUHE-ps from SNNPR noted (P4). Others revealed concern about the quality of training for NCDs. 'I don't think HEWs receive adequate training on NCD services', a health centre participant from SNNPR conceded (P9). A *woreda* health office participant from the same region (P25) considered NCD training was, at best, only an add-on consideration:

> Frankly speaking, we didn't educate them [HEWs] sufficiently about NCDs. What we offered them was no more than a brief orientation. For example, no one from the woreda or regional health bureau came to train them.

Other participants attributed the inattention to NCDs in IRT to the trainers' inadequate knowledge of the subject matter. A *woreda* health office participant from Tigray (P24) stated that:

> Honestly, NCD contents were not covered well during IRT; mostly, we gave just a brief introduction. This might be due to the trainers' low preparedness and the short time slot to deliver the contents.

In some instances, the primary motivation for attendance was the incentives offered as a part of the training sessions. One participant (P4) noted that, 'we [HEW] don't receive any incentive for our work. So, we consider training as the only way to get incentive on top of our

regular salary'. As a result, some HEWs repeatedly attended training sessions primarily to access financial incentives, which limited access for other HEWs who may not have received any training.

While noting the challenges, participants suggested the need for more cost-effective, integrated, need-based and innovative approaches to providing training and improving HEWs' capacity. One UHE-p (P4) stressed the importance of peer-to-peer training: 'It would be good if health care workers from the health centre gave us NCD training through peer-to-peer support'. A participant from the FMoH (P3) commented on the need for enhanced digital learning platforms:

> If we want to sustain the program, we must modify our approach. Some aspects of the training could be provided online, assisted by digital technologies/mobile apps or self-learning by providing hard copy modules.

## Theme 2: Moving beyond checklists: Health extension worker supervision practices for non-communicable disease service delivery

**Subtheme 2.1 Supervision process.** Participants noted that, given the federal nature of the country's political system and regional autonomy, some aspects of the supervision process varied between the two study regions, particularly at the PHC level. As participants described, in SNNPR, health posts were supervised directly by a single supervisor assigned by the health centre within which they operated. A *woreda* health office participant from SNNPR (P12) described that: 'supervisor from PHC unit is assigned to a health post to supervise HEWs on a weekly basis'. Conversely,in Tigray, one supervisor covered up to five health posts as part of the direct supervision process of PHC units. Nonetheless, the *woreda* health office in both settings conducted quarterly integrated supportive supervision.

**Subtheme 2.2 A checklist approach to supervision.** A HEW from Tigray (P15) mentioned that, '[supervisors] follow us routinely in person or by phone and ask what we do. They also help us acquire any supplies, medications, or equipment we need for our work'. However, most study participants reported that it was less supportive with irregular supervisory visits, and when they did take place, the emphasis was on a checklist approach with a focus on task completion. They highlighted that miscommunication between supervisors and HEWs and fault-finding attitudes of supervisors were recurring issues that affected their performance and motivation for their work. One HEW from SNNPR stated that no one asked them about what support they needed (P5), while another HEW from Tigray (P16) noted that:

> The supervisors order us rather than guide us. Moreover, they ask us to perform activities that are not part of our work plan or compatible with our role. We are not satisfied with their supervision. It is discouraging.

Moreover, participants questioned supervisors' problem-solving, counselling and supervision capacity. 'for that matter, I am not sure if they [supervisors] can or have the capacity to supervise us?', commented a HEW from Tigray (P18). 'In my opinion, HEP supervisors have lower capacity than us [HEWs]. Hence, they feel challenged to provide proper technical support', another HEW from the same region (P22) remarked. A FMoH (P2) participant acknowledged that:

> Actually, a supervisor should be a mastermind and have complete knowledge about the HEP. But, in practice, I have found supervisors less experienced in various HEWs' core competence areas and most don't know the HEP manual.

**Subtheme 2.3 Supervisor capacity for non-communicable diseases.** In addition to the above nonexclusive issues related to supervision, participants also noted specific challenges that were unique to the NCD program. Participants described the lack of integration of NCD topics during supervision routines. A HEW from SNNPR (P5) noted, 'I don't remember anyone conducting supervision on NCDs'.

Although some participants reported that supervision checklists did include an NCD component, supervisors may have lackedthe ability to assess service availability and gaps in NCDs service delivery. A *woreda* health official (P24) noted that, 'Actually, NCD is included in the supervision checklist. However, there is often no further discussion or support because the supervisors don't have adequate knowledge of NCDs'.

This highlights the limited availability of NCD training for HEP supervisors. 'The *woreda* staff don't get any refresher training on NCDs. . ..I didn't get any training on NCDs', said a*woreda* health office participant (P11). FMoH officials noted this problem and were looking into providing career advancement opportunities to enable high-performing HEWs to become supervisors.

> We [high-level officials] understand the gap, and we have recently assigned a few degree-holder HEWs as HEP supervisors. And, we hope they [HEWs] will provide the necessary support and solve the problems because they know the HEP and its challenges better". (FMoH participant P1)

## Theme 3: Performance appraisal of health extension workers: Process, enablers and implementation challenges for non-communicable service delivery

**Subtheme 3.1 Health extension worker performance appraisal: Process and implications.** Participants reported that various bodies regularly assessed HEW performance. A health centre participant from SNNPR (P10) observed that:

> HEWs are evaluated by health centres every month. For example, as a HEP coordinator, I evaluate them every two months, and the woreda health office also evaluates them every three months. Finally, their performance is reported to the regional health bureau every three months.

A *woreda* health officer (P24) articulated the critical importance of these performance appraisal reports: 'Every benefit HEWs receive, including training opportunities, is based on their performance evaluation report'. A HEW interviewee (P15) added that 'We need to score 76% or above to keep our jobs. We get fired if we score below the acceptable mark'.

Despite the central importance of performance appraisal, most HEWs felt that it encompassed too many indicators leading to excessive complexity, that was burdensome to HEWs, and ultimately unachievable on the ground. The breadth of activities associated with the HEP and HEW involvement in non-HEP activities meant that most HEWs tended to focus on whatever program received the greatest attention from their supervisors. A HEW from Tigray (P15) observed that this process was unfair:

> The criteria are too cumbersome, and other factors beyond our control, such as government subsidies for constructing toilet facilities and other infrastructure, which we have no control over, also influence our performance score. Very unfair, indeed.

These factors have substantial implications for how HEW performance is assessed and show a high degree of variation across the health system. For example, a HEW from Tigray (P5) indicated that: 'The criteria used to measure HEW performance in each cluster are different, so there is a difference in the evaluation procedure'. A few HEW participants revealed that there was informality in the assessment process, and sometimes it tended to be influenced by personal interactions/interests that the HEWs had with supervisors. Some also complained about the absence of transparency in the process. A HEW from Tigray (P16) highlighted that, 'It wouldn't have any problem if he [the supervisor] have evaluated me openly; then I would have approved and signed'.

Performance evaluators tended to focus on fault finding, and such a work arrangement led to demotivation and low performance among UHE-ps. One participant (P25) argued that as a way forward, 'The performance monitoring framework for HEWs needs to be clear. They should also work without any undue external interference'.'

In addition to the above common gaps associated with HEWs performance appraisal in the HEP as a whole, participants described unique challenges related to the NCD program. The absence of NCD indicators in the performance assessment matrix and the limited attention accorded to NCD during the assessment often pushed HEWs to prioritise programs other than NCDs. A participant from a *woreda* health office (P12) remarked, 'Our key performance indicators assessment matrix does not capture NCDs.'

**Subtheme 3.2 Increased complexity for performance appraisal in urban areas.** Performance appraisal processes were complex for UHE-ps and often lacked clarity in regard to accountability mechanisms. UHE-ps reported that their supervision and performance appraisal might be conducted by the *kebele* administrator, the health centre director, or the immediate supervisor. *Kebele* administrators usually engaged HE-ps in non-HEP related activities, but these activities did not count towards their performance assessment matrix being outside their role, scope of duties and job description. However, performing those tasks for the administration may help them maintain positive relationships with the *Kebele* administration. As participants mentioned, the system consisting of various bodies with different interests managing workers was overly challenging to their performance. This could lead to unrealistic expectations, scapegoating of UHE-ps when the program is not performing, and demotivation of staff. One UHE-p from SNNPR (P6) commented:

Rather than appreciating us, the supervisors focus on picking only our few weak sides. But there should be both appreciation and punishment. Otherwise, it demotivates us. There is no support while we do our job; the leaders just criticise us.

**Subtheme 3.3 Conflicting priorities with non health extension program activities.**
Most participants revealed that HEWs were regularly involved in non-HEP activities, such as engagement in local administrative or political activities, or promotion of non-HEP-related government initiatives. These activities, they argued, increased their workload and compromised HEP service delivery. A health centre participant from SNNPR (P9) described the multiple roles played by HEWs:

She [HEW] has many roles. For example, she is a kabine [35] member of the *kebele* administration; she is a member of the local women and child affairs committee and involved in political activities.

This claim was supported by a senior FMoH official (P3): 'Even when a new program not related to HEP is introduced in a community, they[HEWs] are the ones who go house to

house to promote uptake and its implementation'. 'For this reason, HEWs are not productive', concluded a participant from SNNPR (P9). Despite obvious competing roles, a RHB participant (P27) felt the onus was on the HEW to be more focused:

> Although it is impossible to remove HEWs from political activities, as much as possible, she [HEWs] has to focus on her profession and be engaged in her core role. This is what is being promoted in the office currently.

However, there were mixed opinions from high-level policymakers. Some argued that engagement in non-HEP activities could be empowering and may help HEWs gain community acceptance and the power to bring about change in the community. A participant from the Tigray Regional Health Bureau (P26) noted:

> Their [HEWs] political involvement gives them power. If they don't engage in the local political system [i.e. not being part of the local administration], it may be difficult for them to mobilise the community for different programs.

This was strongly refuted by one HEW from Tigray (P15) who insisted that 'There is no political influence in our job in this *tabiya* [village], but it may be the case in other *tabiyas*'.

## Discussion

Ethiopia's HEP has been a cornerstone of the country's health sector strategy and a critical instrument for expanding essential health services to hard-to-reach areas and improve population health [20, 23]. The success of CHW programs in Ethiopia and elsewhere has been linked mainly to the availability and quality of training, supervision practices, and performance appraisal [6, 15, 36]. In this study, we assessed how such practices had facilitated or hindered HEWs in providing NCD services as part of the HEP. While we found that training, supervision and performance appraisal systems exist within the HEP, most study participants highlighted major limitations in their implementation.Here, we discuss the implications of the findings for policy and practice in Ethiopia and other similar settings. NCD training for HEWs lacked breadth and depth–it was inconsistently delivered with variable availability within and between regionsWhen available, the quality was low with scant content specific to NCDs. Supervision was inconsistent and, rather than supportive, was mainly focused on finding faults in HEW work practices. Supervisors themselves had skill gaps overall and specifically concerning NCDs. HEWs' performance appraisal encompassed too many indicators and was influenced by non-HEP activities leading to excessive complexity that was burdensome to HEWs. Our study identified that while some limitations on training, supervision and performance appraisal are specific to the NCD program, most challenges and barriers tend to overlap with those affecting other HEP services. This reinforces the view that a weak HEP is bad for NCDs and demonstrate the interdependence in support system between NCD and other HEPs.

The limitations we observed in HEW training for NCDs are similar to findings from a recent Ethiopian government report that acknowledges that HEWs continue to have suboptimal knowledge and skill to provide other HEP services. Factors that contributed to the knowledge and skill gap of HEWs, included lack of basic training materials and trainers during pre-service training, training time of insufficient duration, and limited access to in-service training [6]. We found a pressing need for continued support for Level IV trainees as a return to the status quo was contrary to HEW expectations, leading to a demotivated HEW workforce. This was highlighted in the government report, which also noted that in-service training alone

would not address all HEW capacity and motivation gaps [6]. While initiatives such as revising the pre-service training curriculum, better-equipping training institutions and improving practical placements will have some positive results. The challenges identified in our study suggest the need for a more holistic and innovative approach. Such an approach also needs to be complemented by clear career progression opportunities for those participating in these formal training programs.

Our findings highlight the need for IRT content and delivery to be revised and tailored towards addressing HEW capacity gaps and training requirements for major NCDs. Current IRT programs do not reach all regions, and *woredas* and study participants indicated that the irregularity and quality of the training, coupled with the high facilitation costs, had limited their ability to provide sufficient support. These findings are similar to those of a previous study where refresher training on maternal health in Ethiopia failed to meet its aim of improving service delivery [37]. This suggests that the limitations of IRT are entrenched and exacerbated when adding new disease focus areas. Training needs to be delivered more frequently, supporting a lifelong learning approach that continually updates skills [6]. This requires the introduction of a continuing professional development scheme for HEWs with regular revision and adaptation of on-the-job training contents based on the country's evolving disease burden and community health needs. Government initiatives for continuous professional development available for other health care workers through accredited trainings courses and providers should be extended to HEWs. Furthermore, ongoing professional development programs need to be monitored and routinely evaluated for impact on service delivery and identification of future areas of improvement [38, 39].

There are substantial opportunities to digitise training content and deliver interactive online training modules that support continuous learning in more accessible formats. Strengthening technology-assisted training (including via mobile phones) has the potential to improve service delivery, decrease delivery costs and expand the geographic reach of IRT programs, as it eliminates the need for relocation of HEWs for in-person training [38, 40]. Strengthening the link between the health posts and the PHC unit is needed to create and harness self-learning and peer-to-peer learning opportunities for HEWs and promote career progression. Stronger ties with the PHC unit will allow continuous engagement and reduce the costs incurred for formal in-service training. On-job mentoring/coaching and regular seminar sessions organised by health centre staff, such as nurses and doctors, will facilitate continued learning.

Apart from training and capacity development, supportive supervision is considered one of the essential enablers for a satisfied and high-performing health workforce. In Ethiopia, supervision of HEWs is seen as a crucial instrument to ensure quality within the HEP and is emphasised by different levels in the health system [6, 41]. However, our study found that supervision was authoritative rather than supportive, infrequently provided and exhibited wide regional variation in practice. There were significant capacity gaps among supervisors, particularly in mentoring and equipping HEWs with problem-solving skills both overall and for NCDs [18, 36, 42]. Together these factors led to a distinct lack of mentorship in favour of a focus on fault-finding and auditing according to prescriptive and complex checklists. Previous studies have found that inconsistent, poorly organised, and unsupportive supervision can hinder trust and reduce health workers' motivation and performance [18, 36–38, 42]. Supportive supervision which focuses on strengthened relationships, teamwork for the identification and resolution of problems, cross-learning, skill-sharing and constructive feedback has been shown to be effective in improving health worker performance and motivation, and increasing and sustaining job satisfaction in both Ethiopia and elsewhere [18, 41–43]. The WHO recommends developing supervisor training modules that cultivate supportive management practices and enhance

supervisors' problemsolving, coaching and mentoring skills [19]. Continued learning and capacity-building strategies must prioritise empowering HEP supervisors to provide more quality supervision. One promising approach is to create a pathway for Level IV HEWs at a degree program level enabling them to become future supervisors for lower-level HEWs.

Finally, our study revealed major challenges with the performance appraisal process for HEWs. Performance monitoring was considered unnecessarily complex and was often under-pinned by unrealistic targets that were beyond the control of most HEWs. The variability in the performance assessment matrix, and lack of transparency and consistency in the application are key priorities for reform of these programs. Job descriptions with a clear and demar-cated role and tasks are the starting point for implementing an effective performance review process [37]. The performance appraisal system should also have clear and achievable targets and a development component that is time bounded and consistent across job levels. This requires a performance matrix to be more closely aligned with HEP packages and contextua-lised, depending on the resources available in the health posts, and the role and skill sets of the HEW. Overall, but particularly for UHE-ps, clarity on roles and accountability mechanisms is required to improve the performance and motivation of urban workers, particularly when there are competing interests with non-HEP activities that may reflect political or other priori-ties. Nonetheless, it is important to balance HEWs' HEP roles and other non-HEP engage-ments, as a link with *kebele* administrators and other local governors is important as it enables HEWs to obtain political support and access to decision-making processes. With greater clar-ity and transparency around performance appraisal, there is then the potential to implement effective performance-based financial incentives [6].

This study is not without limitations. It was conducted in only two regions, so the findings may not be generalisable to other areas with different contexts. However, the involvement of participants from the FMoH may serve as a moderating voice and balance the views and expe-riences in other regions. Second, the data used in our study were collected in 2019, but much has since changed in Ethiopia. The COVID-19 pandemic, the war in northern Ethiopia and instability in many parts of the country are likely to have shifted government funding priorities and the country's health service needs. In many affected areas, mental health, and other adverse health and health system impacts of the conflict may emerge as significant challenges for the country, and this may, in turn, have substantial implications for the HEP, including for the future role of HEWs moving forward.

The war in Tigray, one of the focus regions for this study, has caused considerable casualties and massive internal displacement. The conflict has also disrupted the HEP and led to a com-plete collapse of the health system in Tigray [44]. At the time of this study, all health posts in the region were fully functional. However a 2021 report from the WHO, indicated that less than half of the health facilities in Tigray were operational at the height of the conflict [45]. Currently, none of the health posts is functional [46]. This will clearly have a devastating impact on healthcare delivery in Tigray. Restoring destroyed health facilities and service deliv-ery disrupted by the conflict may need to go hand in hand with fundamental reorientation of the health system. This may require taking lesson from the pre-war setting (which our study results highlight) but also redefining essential health service packages to meet new needs and address already identified gaps to create a sustainable and resilient health system [47].

## Conclusion

Training, supervision and performance appraisal are key workforce support pillars for any health system. Our found that the training offered to Ethiopian HEWs was irregular and low in quality. These findings appeared to overlap with all the country's HEP focus areas but were

exacerbated when looking specifically at NCDs. Supervision practices were inconsistent, lacked support and mentorship components, and often focused on undertaking auditing activities and findingshortcomings. The HEW performance appraisal was complex, underpinned by high and unattainable targets, and compromised by competing interests with non-HEP activities. To address these challenges, we recommend strategies to enhance pre-service training quality, empower HEWs through continuous professional development programs with enhanced NCD content, and strengthen peer-to-peer learning between staff in health posts and the PHC units. A shift to supportive supervision focusing on collective problem-solving, coaching, and mentoring skills is also needed to accompany training reforms. Performance appraisal should also be reformed to establish clear, transparent and achievable targets that reflect the goals of the HEP packages overall, and specifically those related to NCDs. As some of the limitations on training, supervision and performance also overlap with those of other HEP services, our findings reinforce the view that a robust HEP support system overall would be beneficial for both NCDs and other health conditions.

## Supporting information

**S1 File. English version qualitative interview guides.**
(DOCX)

**S2 File. Consolidated criteria for reporting qualitative research (COREQ) checklist.**
(PDF)

## Acknowledgments

The authors thank the respondents who participated in the interviews. The authors also extend their acknowledgement to the research assistants who dedicated their time and expertise to this project.

## Author Contributions

**Conceptualization:** Azeb Gebresilassie Tesema, David Peiris, Seye Abimbola, Padmanesan Narasimhan, Rohina Joshi.

**Data curation:** Azeb Gebresilassie Tesema, Afework Mulugeta.

**Formal analysis:** Azeb Gebresilassie Tesema, David Peiris, Seye Abimbola, Whenayon S. Ajisegiri, Rohina Joshi.

**Funding acquisition:** Rohina Joshi.

**Investigation:** Azeb Gebresilassie Tesema, Afework Mulugeta.

**Methodology:** Azeb Gebresilassie Tesema, Afework Mulugeta, Rohina Joshi.

**Project administration:** Azeb Gebresilassie Tesema, David Peiris, Seye Abimbola, Rohina Joshi.

**Software:** Azeb Gebresilassie Tesema.

**Supervision:** David Peiris, Seye Abimbola, Rohina Joshi.

**Validation:** David Peiris, Seye Abimbola, Rohina Joshi.

**Visualization:** Azeb Gebresilassie Tesema.

**Writing – original draft:** Azeb Gebresilassie Tesema.

**Writing – review & editing:** Azeb Gebresilassie Tesema, David Peiris, Seye Abimbola, Whenayon S. Ajisegiri, Padmanesan Narasimhan, Afework Mulugeta, Rohina Joshi.

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
