## [Decision Letter · Decision Letter 0]

17 Aug 2022

PGPH-D-22-00922

The influence of health extension workers’ training and supervision on their delivery of non-communicable disease services in Ethiopia

Dear Dr. Tesema,

Thank you for submitting your manuscript to PLOS Global Public Health. After careful consideration, we feel that it has merit but does not fully meet PLOS Global Public Health’s publication criteria as it currently stands. Therefore, we invite you to submit a revised version of the manuscript that addresses the points raised during the review process.

We look forward to receiving your revised manuscript.

Kind regards,

Dinesh Neupane, Phd

Academic Editor

Journal Requirements:

1. Please update your online Competing Interests statement. If you have no competing interests to declare, please state: “The authors have declared that no competing interests exist.”

2. Please amend your online detailed Financial Disclosure statement. This is published with the article. It must therefore be completed in full sentences and contain the exact wording you wish to be published.

Please state what role the funders took in the study. If the funders had no role in your study, please state: “The funders had no role in study design, data collection and analysis, decision to publish, or preparation of the manuscript.”

3. Please ensure that the funders and grant numbers match between the Financial Disclosure field and the Funding Information tab in your submission form. Note that the funders must be provided in the same order in both places as well.

Additional Editor Comments (if provided):

Reviewers' comments:

Reviewer's Responses to Questions

**Comments to the Author**

1. Does this manuscript meet PLOS Global Public Health’s publication criteria? Is the manuscript technically sound, and do the data support the conclusions? The manuscript must describe methodologically and ethically rigorous research with conclusions that are appropriately drawn based on the data presented.

Reviewer #1: Yes

Reviewer #2: Yes

Reviewer #3: Partly

2. Has the statistical analysis been performed appropriately and rigorously?

Reviewer #1: N/A

Reviewer #2: Yes

Reviewer #3: N/A

3. Have the authors made all data underlying the findings in their manuscript fully available (please refer to the Data Availability Statement at the start of the manuscript PDF file)?

Reviewer #1: No

Reviewer #2: Yes

Reviewer #3: Yes

4. Is the manuscript presented in an intelligible fashion and written in standard English?

Reviewer #1: Yes

Reviewer #2: Yes

Reviewer #3: Yes

5. Review Comments to the Author

Reviewer #1: I read this manuscript with interest and found it really relevant and key especially in the context of NCDs. Given the workload experienced by clinicians in SSA in managing patients with chronic diseases, I found the HEP and the role of HEW to be very important work that is being implemented in Ethiopia. I really like the authors effort in Box 1: The context: The Ethiopian health extension program. This work may be useful to the readers of Plos GPH and others actors such as clinicians, researchers or stakeholders working in the field of chronic diseases.

Introduction: Authors can highlight a bot more on why its important to have HEWs in the context of health professional shortage in Ethiopia.

The methods section is well presented. I have added few comments on the PDF on areas of improvement.

In the results sections, I was hoping to get more insights on the current roles of HEWs even before talking about their expanded roles after training. I like the way rural and urban areas have been compared and contrasted in the results sections. However, authors can expound more on the similarities and differences. For example, authors mentioned that UHE-ps were more active or performed more than their rural counterparts. Can the authors explain further why this was the case. For instance, we also know that in urban areas, people are more sensitized than in rural areas, which may make the UHE-ps work easier. In addition, cases of NCDs may be more in urban centres than in rural areas.

I like the discussion. However, authors can provide a paragraph on the research and policy implications of their work.

Manuscript style and format: My main problem with this manuscript is on punctuation marks. Authors really did poorly in formatting this important work. Punctuation are in many places wrongly placed. Authors use both double and single quotation marks in verbatim, etc. Authors need to read through the manuscript and correct this.

See my other comments in the attached PDF.

Reviewer #2: This is a well presented and clear analysis. I only have a few minor comments

1. The authors state the regions and HEWs were selected purposely. Could the authors please expand on this - what were the factors that informed this selection?

2. Could the authors add a box summarising some tentative recommendations based on their findings? Are there any particular lessons that other countries should take note of?

3. Please make use of the COREQ (or equivalent) checklist and include as a supplement - https://www.equator-network.org/reporting-guidelines/coreq/

4. There were several 'local research assistants'. Should they be included in the authorship?

Dmitri Nepogodiev

University of Birmingham, UK

Reviewer #3: This study is very unique in a number of ways; firstly it seeks to appraise and report the extent to which responsiveness to training and supervision of health extension workers (HEW) enhance and support needed skill-building for NCD service delivered by these category of health works in Ethiopia's health extension program (HEP), as suggested by the recommended revised title; "Training and supervision of Health Extension Workers in the delivery of non-communicable disease services in Ethiopia: Outcome of a Qualitative Study". Beyond this, there is the argument that the establishment of the cadre of healthcare workers has rationale underpinning it and is justified. The later argument can only be sustained for policy reform if it is based on sound theoretical and conceptual framework which is absent in the paper. When adequately written, it has policy implications to fully recognize Ethiopia's health extension program and provide the equity it deserves.

However, the following are areas in the manuscript that needs to be strengthened to give it a clear and justifiable argument.

THE TITLE:

The title for this study appears inappropriate for a qualitative research design. The use of the word "influence" gives it an intervention profile whereas it is still an observational study. Therefore a suggestion for modification to align adequately with the design would be; "Training and supervision of Health Extension Workers in the delivery of non-communicable disease services in Ethiopia: Outcome of a Qualitative Study"

ABSTRACT:

The abstract does not adequately represent the study. It lacks information about the problem warranting the study and clear statement of objective, methodology, therefore, as it is expected, the abstract should consist of four distinct paragraphs; it is recommended that the abstract be modified to align adequately.

"Background: The shortage of skilled health workforce and the rising morbidity and mortality from non-communicable diseases (NCDs) in resource-limited settings, such as Ethiopia and elsewhere, have made community health workers' (CHW) programs an essential strategy to address the growing NCD burden. However, the challenge of a growing disease burden at the primary prevention level requires skill-building responsiveness from the respective health systems. In the context of increasing disease burden, this study sought to appraise and report the extent to which responsiveness to training and supervision of health extension workers (HEW) supported needed skill-building for NCD service delivery in Ethiopia's health extension program (HEP).

Methods: A qualitative key-informant in-depth interview (KII) study design was conducted in 2019 in two regions of Ethiopia, the Tigray Region and the South Nations, Nationalities and Peoples Region (SNNPR), and selected 27 consenting policy-makers, district health officials, health center representative and HEW from the Federal Ministry of Health at the federal, regional, district levels who participated. Interview guides were developed guided by the context of the health system responsiveness to the rising NCD burden, recognition of the role of HEW in addressing NCDs, health system strengthening through training and supervision, and principles of primary prevention and modes of intervention..."

BACKGROUND:

It would be observed that the narrative in the background does not specifically set an argument effectively defining the underpinning problem phenomenon. The principles for introducing the HEP is based on Health promotion and disease prevention, which are purely public health principles, thus requires theoretical and conceptual clarifications as part of the review required to establish the scientific foundation.

The last sentence stating the objective of the study mentions "...vis-a-vis their role in NCD prevention in Ethiopia's HEP." This is premised on the health promotion principles at the primary level of disease prevention and control contextualized in the epidemiological principles of the natural history of disease pathogenesis so that their function as HEW are not confused with the clinical function of the physicians and create a conflict needs to be clearly delineated in the literature defining their relevance. This theoretical and conceptual clarification needs to be included in Box 1 for purposes of establishing clarity of the basis of introducing this aspect of the health workforce, as complementary and not duplication, because these are not clinicians.

Another function in the control of NCDs that HEWs may be involved with would be ensuring, by patient education and counselling that the individuals already diagnosed with an NCD and medications prescribed for treatment by qualified medical officer, that they learn to take their medication as prescribed.

This the theoretical and conceptual framework considered at this point provides the public health scientific basis for the argument in the study that "HEW serve to facilitate implementation of primary prevention of Non-communicable diseases that follow a defined epidemiological trajectory of the natural history of disease pathogenesis elucidated by symptomatology of NCDs. The prevention functions of the HEWs is essentially within the primary level of prevention with the two strategic modes of interventions, health promotion and specific protection.

The BOX in the background to the study: There is no basis for creating Box 1 rather this would have been part of the major narrative of the background but elucidating the theoretical and conceptual basis of establishing the HEP.

METHODS

The methodological approach must be consistent with what is stated in the abstract. The study design was "key-informant in-depth qualitative interviews". Reflect this in both the abstract and methodology sub-section of the report.

Under "Participant recruitment and data collection"

The statement "HEWs from the above areas were purposively invited to participate in interviews." is not complete. It should include the justification for the purposive nature of the selection, ie. "HEWs from the above areas were purposively invited to participate in interviews because they are community-based health workers who have experience to share". The basis of purposive sampling is that the sampled have direct relevance to the context and not just any random persons not related to the subject of consideration.

Developing the interview guide: The statement "Interview guides were developed after reviewing different national strategic documents and literature from other settings…" should clarify what specific issues were considered as constituting basis for items of the interview guide. It would have been very appropriate to include the theoretical and conceptual framework as the literature basis.

For emphasis sake; the theoretical and conceptual clarifications constitutes aspects of the literature review contents mentioned above, but this is not obvious in the manuscript at the moment.

RESULTS

The table 1 should have a modified title "Table 1. Characteristics of study participants"

There should also be indication of the gender, length of work experience and possible level of training of participants.

For the results presenting the context, to establish coherence with the public health principles of prevention practices for which this cadres of healthcare workers are established, the recommendation of clarity of framework is necessary. With the theoretical premise elucidated, results should align adequately for consistency and coherence. The context needs minor revision to establish alignment. The first sentence "...the newly revised HEP encompasses the prevention, promotion, early screening of NCDs…". The role of the HEWs is within the primary level of prevention and the corresponding modes of intervention which are health promotion and specific protection. Please refer to the WHO definition of health promotion.

We shall note here that health promotion has three components of Health Education which these HEW are trained to perform, Service strengthening trough training and Policy advocacy. It would be safe to define the limits of function.

The results having the memorable quotes illustrating lessons learned are appropriate.

DISCUSSION

It would have been very appropriate within the first paragraph of the discussion section to reiterate the objective of the study.

The second paragraph begins with "Training for NCDs lacked breadth and depth -it was …" this is not clear. Whose training is being mentioned here?

In paragraph three the statement "The limitations we observed in HEW training for NCDs are similar to findings from a recent government report that acknowledges that HEWs continue to have suboptimal knowledge and skill to provide other HEP services…" did not provide reasons for the observations. This is very important and omission of a reason weakens the argument in the study.

The conclusions should have considered policy statement regarding the establishment of the HEP to establish consistency is aspects considered in the study or lack of it.

6. PLOS authors have the option to publish the peer review history of their article (what does this mean?). If published, this will include your full peer review and any attached files.

**Do you want your identity to be public for this peer review?** For information about this choice, including consent withdrawal, please see our Privacy Policy.

Reviewer #1: **Yes: **Edna N Bosire

Reviewer #2: No

Reviewer #3: **Yes: **Nnodimele Onuigbo ATULOMAH PhD

---

## [Editor Report · Decision Letter 1]

11 Oct 2022

Community health extension workers’ training and supervision in Ethiopia: exploring impact and implementation challenges for non-communicable disease service delivery

PGPH-D-22-00922R1

Dear Mrs. Tesema,

We are pleased to inform you that your manuscript 'Community health extension workers’ training and supervision in Ethiopia: exploring impact and implementation challenges for non-communicable disease service delivery' has been provisionally accepted for publication in PLOS Global Public Health.

Best regards,

Dinesh Neupane, Phd

Academic Editor